# Research Progress in the Biosynthetic Mechanisms of Marine Polyether Toxins

**DOI:** 10.3390/md17100594

**Published:** 2019-10-22

**Authors:** Xiukun Wan, Ge Yao, Yanli Liu, Jisheng Chen, Hui Jiang

**Affiliations:** State Key Laboratory of NBC Protection for Civilian, Beijing 102205, China; xiukunwan@126.com (X.W.); bzyaoge@163.com (G.Y.); liuh306@hotmail.com (Y.L.); chenjsh@cae.cn (J.C.)

**Keywords:** marine polyether toxins, biosynthetic mechanisms, carbon skeleton deletion, pendant alkylation, polyether ring formation, biosynthesis-related genes

## Abstract

Marine polyether toxins, mainly produced by marine dinoflagellates, are novel, complex, and diverse natural products with extensive toxicological and pharmacological effects. Owing to their harmful effects during outbreaks of marine red tides, as well as their potential value for the development of new drugs, marine polyether toxins have been extensively studied, in terms of toxicology, pharmacology, detection, and analysis, structural identification, as well as their biosynthetic mechanisms. Although the biosynthetic mechanisms of marine polyether toxins are still unclear, certain progress has been made. In this review, research progress and current knowledge on the biosynthetic mechanisms of polyether toxins are summarized, including the mechanisms of carbon skeleton deletion, pendant alkylation, and polyether ring formation, along with providing a summary of mined biosynthesis-related genes. Finally, future research directions and applications of marine polyether toxins are discussed.

## 1. Introduction

Marine polyether toxins, mainly produced by dinoflagellates, are natural polyketide compounds with novel and complex structures, unique biological activities, and extensive pharmacological effects, which have attracted great interest among marine biologists and pharmacologists [1,2]. The most striking feature of marine polyether toxins is the presence of one or more ether rings in their molecules, with ring sizes ranging from five- to nine-membered rings. According to their ether ring arrangement and molecular structure, marine polyether toxins are divided into linear, ladder-like (or fused), and macrolide polyethers, the typical examples of which are palytoxins (PLTXs), okadaic acid (OA), pectenotoxins (PTXs), maitotoxins (MTXs), brevetoxins (BTXs), ciguatoxins (CTXs), spirolides, gymnodimines, and pinnatoxins [3,4].

Marine polyether toxins are usually associated with harmful algal blooms, causing severe adverse effects on human health, tourism, wild fish stocks, and aquaculture [4,5,6,7]. For example, BTXs were isolated from the red tide-forming dinoflagellate *Karenia brevis*. The compound can then be enriched in fish, shellfish, and other organisms through the food chain, leading to a large number of fish and shellfish deaths [8,9]. BTXs function by blocking voltage-sensitive sodium channels (VSSCs), which makes this compound a potent neurotoxin [10,11]. CTXs, originally isolated from dinoflagellate *Gambierdiscus* spp., are the main factor causing ciguatera fish poisoning (CFP). CTXs have been shown to interact with the same VSSCs as BTXs do, and both are considered highly potent neurotoxins [12,13]. CTXs accumulate throughout the food chain up to higher predators, and may ultimately reach human consumers, thus causing neurological, gastrointestinal, and cardiovascular disorders. Estimates of the number of people who suffer from CFP poisoning annually range from 50,000 to 500,000 [14,15,16]. OA, which was isolated from dinoflagellate *Prorocentrum lima* and *Dinophysis* spp., is now widely distributed in coastal seas globally. This toxin can lead to diarrhetic shellfish poisoning upon the ingestion of contaminated shellfish by humans [17,18,19].

In addition to their toxicity and harmfulness, marine polyether toxins show special pharmacological activities, with the potential for new drug development or as tools for studying disease-related signaling pathways. BTXs, as a neuroagonist, was shown to increase the plasticity of neurons, revealing its potential to treat diseases such as apoplexy, neurodegeneration, and mucociliary dysfunction [20]. OA has the inhibitory activity against serine/threonine protein phosphatase and can regulate intracellular signaling pathways, which opens up a possibility for its use against Alzheimer’s disease and other neurodegenerative disorders associated with memory impairment [21,22,23]. In addition, OA is a potent inhibitor of tumorigenesis, causing cell growth inhibition and apoptosis of lung and colon cancer cells, and may thus be an important candidate for anticancer drug screening [24,25]. PTXs have demonstrated significant anti-tumor activity against human lung, colon, and breast cancer cells, and are considered potential chemotherapeutic molecules against p53-mutant type tumors [26,27,28]. 

Therefore, studying and exploring the biosynthetic mechanisms of marine polyether toxins could deepen our understanding of the biogenesis and evolution of these compounds, contribute to the effective monitoring, intervention, and elimination of phycotoxins, as well as lay a foundation for the development of new marine-derived drugs or drug precursors. This review focuses on the latest research progress in carbon skeleton deletion, pendant alkylation, polyether ring formation, and newly discovered genes related to the biosynthesis of marine polyether toxins.

## 2. Carbon Skeleton Deletion

Marine polyether toxins are mostly derived from dinoflagellates. It is generally known that the genomes of dinoflagellates are large and complex, with a large number of introns and redundant sequences, and are thus difficult to sequence and annotate and carry out genetic manipulations. Therefore, the biosynthetic mechanisms of marine polyether toxins have not yet been elucidated [29]. Research on the biosynthetic mechanisms of these compounds has mainly been based on isotope labeling experiments, used to identify pathways, or on transcriptome sequencing, used to discover biosynthetic genes. Fortunately, some progress has been made in related research.

Marine polyether toxins belong to a large family of polyketides, the synthesis of which is catalyzed by polyketide synthases (PKSs). The generalities of polyketide biosynthesis have been extensively reviewed over decades [30,31] and will only be briefly described here. Typically, PKS builds carbon chains in a manner similar to fatty acid synthase (FAS), in which the starting substrate, generally acetyl coenzyme A (acetyl CoA), is extended through a series of sequential Claisen ester condensations with malonyl CoA. The ketosynthase (KS) domain, which performs the condensation reaction between acyl units, along with acyl transferase (AT) and an acyl carrier protein (ACP) forms the core structure of FAS and PKS. Other domains that modify the acyl-units after condensation is dehydratase (DH), enoylreductase (ER), and ketoreductase (KR), which are selectively present or absent in PKS, however essential for FAS. The thioesterase (TE) domain hydrolyzes the polyketide chain from ACP, ultimately releasing the polyketide compound from the megasynthase. So far, three types of PKSs have been described [31]. In type I PKSs (modular), catalytic domains are organized in sequential modules on a single polypeptide (multi-domain protein), where each module contains all required domains for each step and is only used once during polyketide assembly, analogous to FASs in animals and fungi. Type II PKSs consist of multi protein complexes where each catalytic domain is present on a separate peptide and functions as a mono-domain protein in an iterative fashion, analogous to type II FASs in bacteria and plants. Type III PKSs, also known as chalcone synthases, are self-contained homodimeric enzymes where each monomer performs a specific function in an iterative manner without the use of ACP.

Based on ^13^C isotope labeling studies of acetate precursors and determination of their chemical structures, it is obvious that marine polyether compounds are produced in dinoflagellates via polyketide synthesis pathways, with acetate being the main extension unit [32,33,34,35]. However, the polyketide chain is not assembled by successive addition of acetate units as in a typical polyketide, rather that the chain contains both intact and cleaved acetate units where the acetate carboxyl or methyl carbons are deleted. This particular phenomenon seems to be common because it has been observed in every dinoflagellate polyketide studied to date, such as BTXs [32,33,34], OA and its derivatives [36,37,38,39,40], goniodomins [41], amphidinols [42,43], amphidinolides [44,45], yessotoxins [46,47], and spirolides [48] (Figure 1), and so can be regarded as a new paradigm in dinoflagellate polyketide biosynthesis. In addition, there are usually multiple acetate deletion steps occurring in a single dinoflagellate polyether. In BTX biosynthesis, there are 12 deletion steps, with 11 carboxyl carbons of acetate units deleted and one methyl carbon deleted. In OA biosynthesis, there are two carboxyl carbon deletion steps. In yessotoxin biosynthesis, there are up to 15 deletion steps, with all carboxyl carbons deleted (Figure 1). 

### 2.1. Carbon Deletion Mediated by Intermediate Metabolites from the Tricarboxylic Acid (TCA) Cycle

To date, there exist three hypotheses regarding the mechanism of carbon deletion in dinoflagellate polyketides. The first suggests that acetate units in the carbon skeleton are not really cleaved or deleted but are converted to new extension units through other pathways prior to condensation with the polyketide chain. In particular, some intermediate metabolites from the tricarboxylic acid (TCA) cycle, such as succinate and α-ketoglutarate, may participate in the growing polyketide chain [33,34] (Figure 2). It is known that one-step oxidative decarboxylation occurs when acetyl CoA is added in the TCA cycle to form succinate and α-ketoglutarate, which corresponds to the deletion of carboxyl carbon in an acetate unit. The participation of succinate and α-ketoglutarate in the BTX biosynthesis was detected via ^13^C isotope labeling experiments, although the location of these units in the BTX carbon skeleton has not been determined. However, this postulate suffers from the fact that the nascent polyketide chain would have to detach from and then reattach to the PKS enzyme. Moreover, a detailed quantitative evaluation of isotope enrichment during BTX biosynthesis from single- and doubly-labeled acetate established that all carbons, including the deleted ones, were equally enriched [49]. Similar results have also been found for OA biosynthesis [50]. These findings indicated that all acetate building units were derived from the same biogenetic bank, and it seemed unlikely that other biogenetic precursors were involved because this would lead to variations in isotope enrichment along the polyketide chain. 

### 2.2. Carbon Deletion Through Favorskii-Type Rearrangement

The second presumption suggests that carboxyl carbons of intact acetate units are deleted from the nascent polyketide chain through a Favorskii-type rearrangement, catalyzed by a cytochrome P450 enzyme or flavin monooxygenase [50,51] (Figure 3). Briefly, the methyl carbon of acetate unit is first oxidized by a P450 enzyme or flavin monooxygenase to form an α-keto group. Then, the neighboring methyl group attacks the electrophilic keto-carbon and forms a cyclopropanone intermediate, followed by a flavin-derived peroxide reaction and a release of the carboxyl carbon as carbon dioxide. Hence, this rearrangement yields a shortened chain with a deletion of the carboxyl carbon. The Favorskii-type rearrangement, resulting in the excision of a carbon from the polyketide chain, is extremely rare in other polyketides and has only been noted in early labeling studies of the bacterial product enterocin [52], the fungal products aspyrone [53] and asperlactone [54], and the myxobacterial products ambruticin [55] and wailupemycin A [56,57]. The P450 enzyme or monooxygenase mediating Favorskii-type rearrangement in these polyketides have been identified [58,59,60]. In the cases of enterocin, aspyrone, and asperlactone, the carboxyl carbon is not completely deleted from the molecule but is retained as a carboxyl side group, whereas in ambruticin and wailupemycin A, the carboxyl group is actually lost. To date, no analogous isozymes or homologous genes mediating the Favorskii-type rearrangement of marine polyether toxins have been identified in dinoflagellates.

### 2.3. Carbon Deletion via Specific Functional Modules within PKS

According to the third proposed mechanism, the carbon chain is formed exactly as in normal polyketides. During the polyketide elongation process, the terminal β-keto-group is reduced to an alcohol and dehydrated to an α,β-unsaturated acid. Then, the nascent polyketide chain is attached to a special module, which contains three key enzyme domains, TE, epoxidase (EP), and decarboxylase (DC). The polyketide chain is first subjected to TE and detached from PKS, then the double bond undergoes epoxidation, and the resulting glycidic acid structure can undergo facile decarboxylation to form an aldehyde with one less carbon. If the decarboxylation occurs oxidatively, the product is a carboxylic acid with one less carbon, which, after thioesterification, can undergo further elongation [49] (Figure 4). This hypothesis seems reasonable, but it requires a special PKS module. At present, the corresponding EP and DC domains have not been identified. In addition, the extended polyketide chain needs to be dissociated from PKS, then reactivated, and connected to PKS again after decarboxylation. The related reaction and its regulatory mechanism have not yet been studied.

The three hypotheses offer clues and references for studying the dinoflagellate polyketide chain construction mechanism, among which the second proposition involving the Favorskii-type rearrangement seems the most reasonable, as this carbon-deletion process would not interrupt the nascent polyketide chain extension process, nor would it involve the detachment from and subsequent reattachment to the PKS enzyme. Obviously, this process could also take place if the chain has been released from the PKS enzyme. More definitive elucidation must be based on the investigation of genes and enzymes mediating carbon skeleton deletions, such as the FAD-dependent oxygenase, cytochrome P450, or flavin monooxygenase. In addition, more detailed isotope labeling experiments should be carried out to obtain biosynthetic information by precisely tracking the intermediate metabolites.

## 3. Pendant Alkylation

### 3.1. α-Alkylation

In the majority of bacterial and fungal polyketides, pendant methyl groups occur through the incorporation of propionate or the addition of the electrophilic methyl group of S-adenosyl methionine (SAM), catalyzed by the methyl transferase enzyme located in the polyketide biosynthetic gene cluster [61,62]. This reaction is described as α-alkylation because pendant alkylation occurs at a nucleophilic α-carbon in the polyketide chain (Figure 5A). In isotope labeling experiments of polyether compounds derived from dinoflagellates, α-alkylation could also be found. For example, there are four pendant methyl groups derived from methionine in BTX (Figure 1). 

### 3.2. β-Alkylation

In addition to α-alkylation, there is another type of methylation, which occurs at the electrophilic β-carbon of acetate units in the polyketide chain. Isotope labeling experiments showed that pendant methyl groups were derived from the methyl carbon of acetate, and this process is called β-alkylation. Apparently, β-alkylation regularly occurs in dinoflagellate polyethers, as every dinoflagellate polyether studied to date contains at least one β-alkylation site. For example, there are three β-alkylation sites in BTX, four in OA, and two in yessotoxin (Figure 1). In addition to polyether toxins from dinoflagellates, β-alkylation has also been reported for other polyketides, such as myxobacterial compounds myxopyronin A [63] and myxovirescin A [64,65,66], cyanobacterial products curacin [67] and jamaicamide [68], and the bacterial metabolites pederin [69], bacillaene [70,71], difficidin [72], leinamycin [73,74], and mupirocin [75,76]. One β-alkylation step occurs during the synthesis of mupirocin, a metabolite of *Pseudomonas aeruginosa*, through an aldol condensation reaction catalyzed by 3-hydroxy-3-methylglutaryl-CoA (HMG-CoA) synthase, which was identified by gene sequencing and functional analysis. When HMG-CoA synthase is mutated, the β-alkylation reaction cannot occur, and the carbon chain cannot be extended, resulting in a truncated mutant product mupirocin H [75,76]. This finding suggests that HMG-CoA synthase is responsible for catalyzing the β-alkylation reaction. Gene sequencing showed that the gene for HMG-CoA synthase was incorporated into the polyketide biosynthetic gene cluster.

To study the β-alkylation mechanism of OA, all three hydrogen atoms in the acetate methyl carbon were labeled with deuterium (CD_3_). The results showed that only two deuterium isotopes were present in the pendant methyl group of OA. Therefore, it is speculated that acetate is first activated to malonate, and then malonate is condensed with the β-carbonyl site of the new carbon chain through an aldol condensation catalyzed by HMG-CoA synthase (Figure 5B) [50]. Therefore, although there is still no evidence at the molecular and genetic levels, it is reasonable to conclude that a similar mechanism involving HMG-CoA synthase, as found in various bacteria, is responsible for the β-alkylation reaction of dinoflagellate polyethers.

### 3.3. Pseudo α-Alkylation

It is noteworthy that five of the six methyl groups in the side chain of OA are located in the β-carbonyl site and originate from the β-alkylation reaction discussed above, while the last methyl group (C-10 position) occurs at the methyl carbon (α-carbon) of a cleaved acetate unit in the polyketide chain. Isotope labeling experiments indicated that the methyl group was derived from acetate but not from methionine. Therefore, this special type of alkylation is referred to as pseudo α-alkylation. It is presumed that the pseudo α-alkylation reaction proceeds via an acetate unit, which first undergoes the deletion of the carboxyl carbon, followed by the oxidation of the retained, methyl group to a carbonyl group. Finally, similar to the process of β-alkylation, HMG-CoA synthase catalyzes aldol condensation of the carbonyl group with activated malonate to form a methyl side chain [50]. Notably, pseudo α-alkylation is very common in the process of dinoflagellate polyether biosynthesis, and every dinoflagellate polyether studied to date contains at least one pseudo α-alkylation site (Figure 1).

Therefore, β-alkylation (including pseudo α-alkylation) is a common paradigm in dinoflagellate polyketide studies. Analysis of other β-alkylation examples and biosynthetic pathways from various bacteria showed that HMG-CoA synthase is the key enzyme catalyzing the β-alkylation reaction. Thus, transcriptome sequencing, gene probe capture techniques, and bioinformatics analysis are needed to find homologous HMG-CoA synthase genes in dinoflagellates and determine their function.

## 4. Polyether Ring Formation

A distinct feature of dinoflagellate polyether compounds is the presence of a number of five- to nine-membered spiral or fused polyether rings. According to their ether ring arrangement and molecular structure, dinoflagellate-derived compounds can be divided into linear, macrolide, and fused polyethers [3]. Thus, OA belongs to linear polyethers with one five-membered ring and six six-membered rings. BTXs are fused polyethers with eight six-membered rings, two seven-membered rings, and one eight-membered ring. Spirolides are macrolide polyethers with one five-membered ring and five six-membered rings (Figure 1). Based on the unique size, number, and spatial configuration of ether rings in dinoflagellate polyether compounds, which are quite different from other polyether compounds (such as polyether antibiotics), a unique mechanism of ring formation may exist in dinoflagellates.

### 4.1. Baldwin’s Rules for Ether Ring Formation

In order to describe the ring formation mechanism, it is necessary to understand the ring formation rules proposed by Baldwin [77], which states that the number of atoms in the ring is generally three to eight. The terms *exo* and *endo* are used to indicate the position (inside and outside the ring, respectively) of the bond broken during the ring closure in the newly formed ring. The terms *tet* (tetrahedral), *trig* (trigonal), and *dig* (digonal) are used to indicate the geometry of the electrophilic carbon in the ring, with *tet* corresponding to sp^3^ hybridization, *trig* to sp^2^ hybridization, and *dig* to sp hybridization. The rationalization of Baldwin’s ring formation rules usually has a three-dimensional electron effect. Whether the reaction is favored or disfavored is mainly determined by the molecular dynamic equilibrium, and the most advantageous type of ring formation usually follows the principle of minimum free energy. The common types of ring formation are shown in Figure 6. If the breaking position is at a carbonyl group or a double bond, and the latter is outside the ring, the condensation of the intramolecular hydroxyl group with the sp^2^ hybrid carbon forms an *exo-trig* cyclization type, which is usually favored (Figure 6A). When the double bond breaking position is inside the ring, and the ring size is 6-7, the condensation of the intramolecular hydroxyl group with the sp^2^ hybrid carbon represents an *endo-trig* cyclization type reaction, which is usually favored, but is disfavored when the ring size is 3-5 (Figure 6B). If the breaking position is at a hydroxyl group or an epoxide bond, and the latter is outside the ring, the condensation of intramolecular hydroxyl group with the sp^3^ hybrid carbon follows an *exo-tet* cyclization type reaction, which is usually favored (Figure 6C). When the epoxide bond is inside the ring, the condensation of intramolecular hydroxyl group with the sp^3^ hybrid carbon follows an *endo-tet* cyclization type reaction, which is usually disfavored (Figure 6D). Overall, the Baldwin rules for cyclization reasonably explain when polyether ring formation is thermodynamically favored or disfavored. However, enzymes are capable of overcoming kinetic barriers and catalyzing reactions in a thermodynamically disfavored direction.

### 4.2. Examples from Polyether Antibiotics Relevant to Polyether Formation

So far, polyether antibiotics from *Streptomyces*, such as monensin A, nanchangmycin, lasalocid A, and salinomycin, have been found to have the highest structural similarity to dinoflagellate polyether compounds. Polyether antibiotics are catalyzed by PKS and also contain multiple ether rings, providing some reference for studying the mechanisms of dinoflagellate ether ring formation. Based on early isotope labeling experiments, Cane and Westley proposed a possible mechanism for the biosynthesis of polyether antibiotics in 1983 [78,79,80,81]. First, acetic acid, propionic acid, or butyric acid units are selectively condensed by PKS to form acyclic polyene intermediates in the all *E* (*trans*) configuration. Then, epoxides are formed by epoxidation of double bonds, followed by the breakage and re-closure of epoxy bonds to form the characteristic ether bonds and spiral ring structures. After a series of post-modification processes, such as glycosylation, O-methylation, and hydroxylation, mature polyether antibiotics are produced. The hypothesis was later confirmed by successive discovery and analysis of several antibiotic biosynthetic gene clusters and by validation of functional genes [82,83,84,85,86,87,88,89,90]. The key genes responsible for polyether formation were discovered in all the polyether antibiotic biosynthetic gene clusters, among which epoxidases catalyze the conversion of polyene intermediates to epoxides, and epoxide hydrolases are responsible for the opening of epoxy bonds and the formation of ether rings. In the monensin A gene cluster, MonCI has the function of epoxidation, catalyzing epoxidation of the three double bonds of triene intermediates [88,89], while MonBI and MonBII are responsible for catalyzing the hydrolysis of epoxides and the formation of ether rings [91]. The ether rings are formed through *6-exo-trig* and 5-*exo-tet* cyclization reactions, which are favored as discussed above, and conform to Baldwin’s cyclization rules (Figure 7A). In the lasalocid A biosynthetic gene cluster, Lsd18 catalyzes the epoxidation of diene intermediates [87,90], while Lsd19 catalyzes the hydrolysis of epoxides and the formation of ether rings [92]. Among the two ether rings, the first one (C18–C19 epoxy bond) is formed via a 5-*exo-tet* cyclization reaction, which conforms to Baldwin’s cyclization rules and is favored, but the second one (C22–C23 epoxy bond) is formed via a 6-*endo-tet* cyclization reaction, which is disfavored from the chemical energy standpoint, indicating a need for enzyme participation. The Oikawa team [93,94] analyzed the crystal structure of Lsd19 and its substrate binding to elucidate the catalytic mechanism. Lsd19 consists of two homologous domains, namely Lsd19A and Lsd19B. A series of site-directed mutagenesis experiments showed that Lsd19A recognizes the C18–C19 epoxy bond and catalyzes 5-*exo-tet* cyclization at this site. Next, Lsd19B catalyzes the opening of C22–C23 epoxy bond and formation of the six-membered ether ring via a 6-*endo-tet* cyclization reaction. In vitro experiments also confirmed that Lsd19 could catalyze the conversion of epoxidized lasalocid precursors to lasalocid A via 5-*exo-tet* and 6-*endo-tet* cyclization reactions, while in the absence of Lsd19, isolasalocid but not lasalocid A was obtained through two-step 5-*exo-tet* cyclization reactions catalyzed by trichloroacetic acid [92] (Figure 7B). Therefore, epoxide hydrolases could overcome the *endo-tet* cyclization reaction, which is disfavored in terms of the chemical energy, and catalyze the epoxide hydrolysis to form an ether ring.

### 4.3. Ether Ring Formation in OA

Although no experimental data have been reported so far elucidate the functions of epoxidases and epoxide hydrolases in dinoflagellate ether ring formation, several isotope labeling studies indicated the origin of oxygen atoms in the OA carbon skeleton and provided some clues for understanding the mechanism of OA ether ring formation (Figure 8). Needham et al. [39] used ^13^C and ^18^O isotopes to label acetate (1-^13^C, ^18^O_2_) and glycolic acid (2-^13^C, ^18^O), respectively, during OA biosynthesis. The results showed that the oxygens at the C-4 and C-27 positions were derived from acetate, which indicated that the carbonyl group of acetate was retained or reduced to a hydroxyl group during the condensation reaction, while the oxygen at the C-38 position was derived from glycolic acid. Murata et al. [35] used ^18^O isotopes to label acetate (1-^18^O_2_) and free oxygen (^18^O_2_) in OA biosynthesis. The results showed that the oxygens at the C-4, C-8, C-19, and C-27 positions were derived from acetate, while those at the other positions, except C-1 (hydroxyl group) and C-38 (oxygen), were derived from free oxygen (^18^O_2_), suggesting the involvement of hydroxylation or epoxidation reactions. Izumikawa et al. [95] used ^18^O-labeled water (H_2_O-^18^O) in OA biosynthesis and found that the oxygens at the C-1, C-7, C-8, C-19, C-24, C-34, and C-38 positions were labeled, indicating that the oxygens at these locations may have undergone reversible hydration with water and exchange with oxygen in water. Based on these experiments, it can be concluded that oxygens at multiple sites in OA are derived from different sources. For example, the oxygens at C-8 and C-19 positions are derived from both acetate and water. This finding indicates that carbonyl groups are formed at these positions during the extension of an acetate unit, and then the carbonyl groups undergo reversible hydration with water and exchange with oxygen in water. The oxygens at the C-1, C-7, C-24, and C-34 positions are derived from both oxygen and water, indicating that the carbons at the C-1 and C-34 positions may be first oxidized to a carbonyl group, and then reversibly hydrated with water, while the carbons at the C-7 and C-24 positions are oxidized to a hydroxyl group, and free oxygen may be generated by photosynthesis of water. Meanwhile, in addition to being derived from acetate and water, the oxygen at the C-8 position is also derived from the oxygen, which is puzzling, as it is unlikely that free oxygen is needed if the acetate unit has provided a carbonyl group. The C-38 oxygen, derived from glycolic acid, may also come from water, which is difficult to explain. Thus, these results need further experimental verification.

Nevertheless, based on the above results, it is possible to infer the mechanism of each ether ring formation in OA [96]. As shown in Figure 8, based on the intensity of ^18^O isotope and the composition of glycolic acid in the G ring, the ring formation proceeds in the order from G to F ring, and the G ring is the initiation ring. First, the hydroxyl group at the C-38 position attacks the carbonyl group at the C-34 position, forming the G ring via a 6-*exo-trig* cyclization reaction. Then, the F ring is formed via a 6-*endo-trig* cyclization reaction. Both steps are favored in terms of chemical energy. Next, the C, D, and E ring formation proceed in the order from right to left. As the D and E rings are fused rings, and according to isotope labeling experiments, C-19 should be a carbonyl group, while the C-22 and C-23 carbons should form an epoxy bond. First, the hydroxyl group at C-26 position attacks the epoxy bond and forms the E ring via a 6-*endo-tet* cyclization reaction. Then, the hydroxyl group at C-23 position attacks the carbonyl group at C-19 position via a 6-*exo-trig* cyclization reaction to form the D ring. Subsequently, the hydroxyl group at C-19 position attacks the hydroxyl group at C-16 position to form the C ring via a 5-*exo-tet* cyclization reaction. Only 6-*endo-tet* cyclization is disfavored, while the other two steps are favored. Finally, the order of the A and B ring formation should proceed from left to right, contrary to that of the other rings. Based on isotope labeling experiments, the C-4 oxygen is only derived from acetate, indicating the presence of a hydroxyl group at this position, while the C-8 oxygen is derived from both acetate and water, indicating the presence of a carbonyl group at this position, which suggests that the C-4 hydroxyl group attacks the carbonyl group at the C-8 position and forms the A ring via a 6-*exo-trig* cyclization reaction. Then, the C-8 hydroxyl group attacks the C-12 hydroxyl group and forms the B ring via a 6-*exo-tet* cyclization reaction. Both steps are favored in terms of chemical energy. As Figure 8 shows, there are both spiral and fused rings in the carbon skeleton of OA, and their cyclization reactions are separate and different. The spiral ring formation often follows the favored *exo-trig* or *exo-tet* route, while the fused rings are usually formed via *endo-tet* reactions and involve epoxy bonds. In addition, according to the mechanism of ether ring formation in polyether antibiotics, the formation of the OA polyether should also require epoxidases, which is responsible for the formation of epoxy bonds, and epoxide hydrolases, which is responsible for the opening of epoxy bonds and ring formation. Although the structure of OA is relatively simple, the diversity of its ring formation reactions provides a reference for studying the ring formation mechanism of more complex polyether compounds.

### 4.4. Ether Ring Formation in Fused Polyethers

The typical characteristic of more complex, fused polyether compounds is the presence of a series of fused ether rings, showing a typical *syn-trans* configuration. It is presumed that these ether rings are formed by continuous *endo-tet* cyclization of *trans* polyepoxides, which are produced through oxidization of all-*E* polyene precursors [97,98] (Figure 9A). For example, CTX has 11 continuous fused rings (A to K) and two spiral rings (L and M). It is presumed that polyether rings are first condensed by PKS and then modified through multiple steps to form a polyketide polyene precursor. Except for the first double bond, the other double bonds in the *E* configuration are all oxidized to form all-*trans* polyepoxides catalyzed by epoxidases. The polyepoxide intermediates are then cyclized by epoxide hydrolase to form 11 consecutive fused rings via successive *endo-tet* cyclization reactions, followed by 6-*exo-trig* and 5-*exo-tet* reactions to form the terminal spiral rings (Figure 9B). The 6-*exo-trig* and 5-*exo-tet* cyclization reactions are favored, while the 11 successive *endo-tet* cyclization reactions are disfavored. Vilotijevic and Jamison [99] found that polyepoxide precursors, containing an initial ether ring, could spontaneously undergo 6-*endo-tet* cascade cyclization in vitro in an aqueous environment with neutral pH to form the fused ring structure. Water acted as a catalyst and possibly stabilized the conformation of the ether ring through hydrogen bonds, thus facilitating the nucleophilic attack of hydroxyl groups via the 6-*endo-tet* route to form the ether rings, with the initiation ether ring providing the template and the hydroxyl group for nucleophilic reactions. This discovery suggests that the water molecule plays a role in promoting the activity of epoxide hydrolase for the formation of polycyclic fused ethers, resulting in the disfavored 6-*endo-tet* cyclization reactions.

Thus, the mechanisms of ether ring formation differ in various dinoflagellate polyether structures, such as linear, macrolide, or fused polyethers. In linear and macrolide polyethers, rings are usually 5- or 6-membered as in polyether antibiotics, and are mainly connected as a spiroketal feature, or separated by at least one single bond [3,78]. Thus, these ether rings are usually formed via the favored *exo* cyclization reaction and conform to Baldwin’s rules. Certainly, epoxidases and epoxide hydrolases may also be needed to catalyze specific reactions, even though these rings can be formed spontaneously from the chemical energy standpoint [88,89,90,91,92]. In the more complex fused polyethers, such as CTXs and BTXs, there are a series of successive fused ether rings, varying in size from 5- to 9-membered rings [3,4]. Thus, fused ether rings are formed via the disfavored *endo-tet* cyclization reaction and must rely on catalytic enzymes, such as epoxide hydrolases. Therefore, discovering and characterizing the functions of these enzymes is essential for the elucidation of the mechanisms of dinoflagellate polyether ring formation.

## 5. Gene Mining 

The above-discussed biosynthetic mechanisms of marine polyether compounds have mostly been proposed based on the results of early isotope labeling studies, in the absence of information on genes and enzymes involved in catalytic reaction steps. As the genome sequences of dinoflagellates are very long and complex, it is difficult to uncover the biosynthesis-related genes. At present, most studies are based on transcriptome sequencing of dinoflagellates. With the rapid development of high-throughput sequencing technologies, the dinoflagellate transcriptome has been increasingly sequenced and annotated, which provides the basis for studying biosynthetic mechanisms.

### 5.1. New Single-Domain Type I PKS

Snyder et al. [100] amplified the KS domain gene sequences of PKS from seven dinoflagellates for the first time, using a degenerate PCR method. Three of the dinoflagellates examined are confirmed polyether producers, including the benthic dinoflagellates *P. lima* and *P. hoffmanianum*, which produce OA, and the planktonic dinoflagellate *K. brevis*, which produces the fused polyether BTXs. The other four strains have not been demonstrated to produce polyketides, yet they belong to polyketide-producing genera. These gene sequences determined are highly homologous to type I PKS gene sequence, as shown by bioinformatic comparative analysis. Subsequently, flow cytometry and fluorescence probe in situ hybridization techniques were used to demonstrate that two of the gene sequences were localized exclusively to *K. brevis*, providing further evidence that the gene sequences were derived from dinoflagellates [101]. Monroe and Van Dolah [102] identified eight full-length PKS gene sequences of *K. brevis* by constructing and sequencing a cDNA library and using expressed sequence tags and rapid amplification of cDNA ends techniques. Six of the sequences were single KS domain sequences, one was KR domain sequence, and the last one encoded both ACP and KS domains. Sequence alignment showed that these sequences were homologous to those of type I PKSs. Despite the sequence homology, the structure of dinoflagellate PKS differs from that of the typical type I PKSs, containing multi domains, but is similar to that of type II PKSs. Therefore, the dinoflagellate PKS characterized by the presence of single-domain structure is called the ‘new type I PKS’. Kohli et al. [103] used second generation high-throughput sequencing technology to sequence and assemble transcriptomes of two dinoflagellates (*Gambierdiscus australes* and *G. belizeanus*) producing MTXs. More than 300 type I PKS-related sequences were identified through sequence alignment analysis, including PKS (KS, KR, ER, AT, and ACP), TE, and methyltransferase genes, all of which encoded single-domain proteins. In addition, other single-domain type I PKS-related genes were identified in other dinoflagellates [104,105,106,107,108], suggesting that type I PKS with a single-domain structure may be a prominent feature of dinoflagellates.

### 5.2. Typical Multi-Domain Type I PKS

In addition to the discovery of type I PKS with a single-domain structure, several studies have found PKSs with multi-domain structure in dinoflagellates. Lopez-Legentil et al. [109] constructed a fosmid genomic library of the dinoflagellate *K. brevis* by nested PCR and cloned 18 gene sequences with the single-KS domain. Furthermore, a multi-domain NRPS-PKS hybridization sequence was identified, of which the NRPS consists of three modules, including a condensation domain, an adenylation domain, and a peptidyl carrier protein domain, while the PKS is composed of KS, AT, KR, ACP, and TE domains, suggesting that the heterozygous gene module may catalyze the synthesis of some amine-containing polyether compounds. Beedessee et al. [110] identified 25 type I PKS genes by genome and transcriptome sequencing analysis of the dinoflagellate *Symbiodinium minutum*, among which 15 were single-domain sequences (KS), and 10 were multi-domain modular PKS sequences, containing one NRPS-PKS hybrid sequence. Subsequently, Kohli et al. [111,112] used the second generation high-throughput sequencing technology to sequence the transcriptomes of *K. brevis*, *G. polynesiensis*, and *G. excentricus*. Using sequence alignment and phylogenetic analysis, hundreds of type I PKS genes were identified, including not only single-domain sequences but also multi-domain sequences. In addition, epoxidases, epoxide hydrolases, and methyltransferases were identified. These results provide a wealth of genetic information for the study of the biosynthetic mechanisms of dinoflagellate polyether compounds. Thus, with the rapid development of high-throughput sequencing technologies and the improvement of gene cloning methods, genes related to polyether biosynthesis, including PKS, are constantly being excavated in dinoflagellates. From the early discovery of genes encoding single-domain PKS to the later discovery of multi-domain modular genes, these results suggest that the structure of PKS in dinoflagellates is characterized by diversity. They have not only the typical structural characteristic of multi-domain modular type I PKS, but also the characteristic of single-domain sequence similar to type II PKS gene, which indicates that the catalytic mechanisms of dinoflagellate PKS are different from those of the two types of PKSs or may combine catalytic mechanisms of both. These findings also illustrate the complexity of the biosynthetic mechanisms of dinoflagellate polyether compounds at the genetic level and the difficulties in analyzing their biosynthetic pathways. Although a large number of PKS genes have been discovered, most of the gene sequences are incomplete, and their functions have not been characterized. The catalytic mechanisms of PKS in biosynthetic pathway are still unclear, making it necessary to study numerous enzymatic reactions in vitro, as well as to develop methods of genetic manipulation for dinoflagellates.

## 6. Prospect 

Owing to their novel structures and unique biological activities, dinoflagellate polyethers have good prospects for drug development. These compounds also include red tide toxins, which pose a major threat to marine fisheries, environment and human health. Therefore, dinoflagellate polyethers have attracted extensive attention from researchers. On the one hand, research on the biosynthetic mechanisms can provide a basis for the development of monitoring technologies or biosensors for this kind of algal toxin. For example, fluorescent probes and gene chips can be designed for in situ, real-time, and high-throughput monitoring of specific gene sequences (KS, epoxide hydrolase, etc.). Real-time fluorescent quantitative PCR or fluorescent antibody, which are sensitive and reliable even at low toxin concentrations, can be used for detection of the expression level of functional enzymes in biosynthetic pathways, so as to give early warning and effective prevention of the growth of the poisonous algae. In addition, blocking methods can be designed for their biosynthetic pathways, such as inhibiting the activity of functional enzymes, silencing or knocking out genes related to biosynthesis, so as to control or even eliminate algae toxin production. On the other hand, studies of dinoflagellate polyethers can lay the foundations for the development of tools to study disease signaling pathways or new marine-derived drugs. Dinoflagellate polyethers can act on various ion channels and play an important role in the nervous system, thus potentially representing powerful tools for pharmacological and neuroscience research. These compounds may also have the potential for the development of new cardiovascular drugs, neurological injury treatment drugs as well as anti-cancer drugs. Through analysis of the biosynthetic mechanisms of dinoflagellate polyethers and the discovery of biosynthesis-related genes, it may be possible to reconstruct their biosynthetic pathways in heterologous hosts using metabolic engineering and synthetic biological methods and also create new compounds by inserting, deleting, or modifying PKS modules and domains. 

Previous isotope labeling studies revealed the origins of carbon and oxygen atoms in the carbon skeleton of several dinoflagellate polyether compounds. Subsequently, numerous PKS genes were identified through gene cloning and transcriptome sequencing, which further confirmed that dinoflagellate polyether compounds were produced through biosynthetic pathways catalyzed by PKS. However, due to the lack of mature genetic manipulation tools, gene function characterization, and genome sequencing, the specific biosynthetic mechanisms of dinoflagellate polyether compounds, including carbon skeleton deletion, pendant alkylation, and polyether ring formation, are still unclear, instead, remain speculations based on their structures and available isotope labeling experimental data. Therefore, it is of prime importance to develop approaches to the genetic manipulation of dinoflagellates, in combination with more advanced sequencing technologies, to allow deeper insights into their genomes, as well as the acquisition of more complete information on biosynthesis-related genes and enzymes, such as monooxygenases, HMG-CoA synthases, epoxidases, epoxide hydrolases, and methyltransferases. At the same time, in vitro experiments should be carried out to identify the genes and characterize the functions of key enzymes or other proteins involved in the biosynthetic pathways. In addition, more information can be learned about the biosynthetic mechanisms of dinoflagellate polyether compounds from studies of other polyketide compounds, especially polyether antibiotics, which will be beneficial for understanding the more complex biosynthetic mechanisms of dinoflagellate polyether compounds.

## Figures and Tables

**Figure 1 marinedrugs-17-00594-f001:**
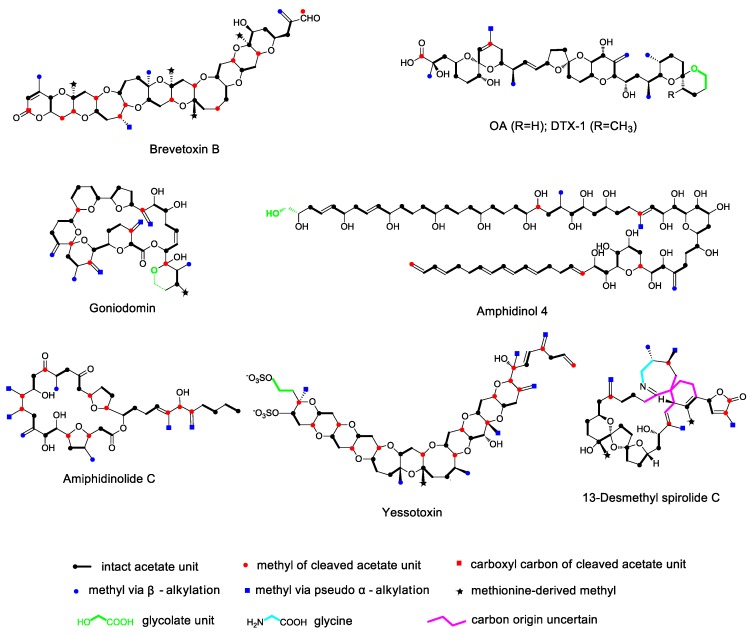
Isotope distribution in some marine polyether toxins following incorporation of ^13^C-labeled acetate, methionine, and glycolate. The carbon backbone contains intact acetate units as well as ones that have been cleaved within the chain. The pendant methyl groups are either derived from S-adenosyl methionine (SAM) or from acetate by the 3-hydroxy-3-methylglutaryl-CoA (HMG-CoA) pathway.

**Figure 2 marinedrugs-17-00594-f002:**
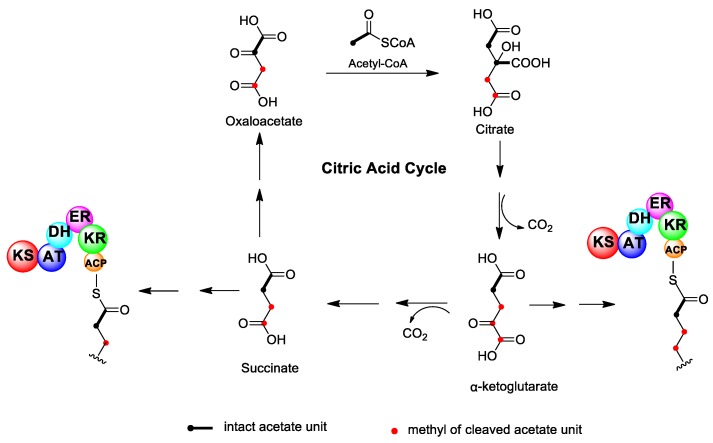
One of the proposed hypotheses for the carbon deletion mechanism in dinoflagellate polyketides. Succinate and α-ketoglutarate, intermediate metabolites from the tricarboxylic acid (TCA) cycle, may participate in the growing polyketide chain.

**Figure 3 marinedrugs-17-00594-f003:**
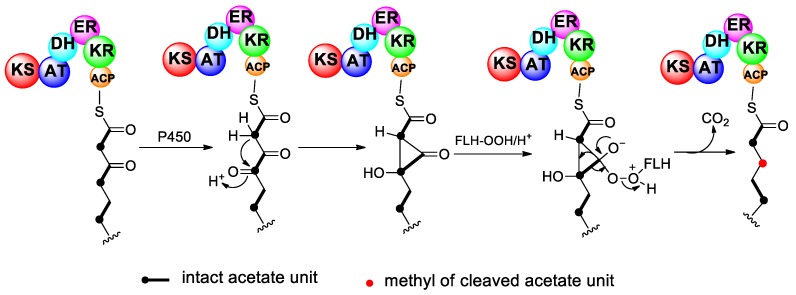
Proposed mechanism for the deletion of the carboxyl carbon from an intact acetate unit via the Favorskii-type rearrangement.

**Figure 4 marinedrugs-17-00594-f004:**
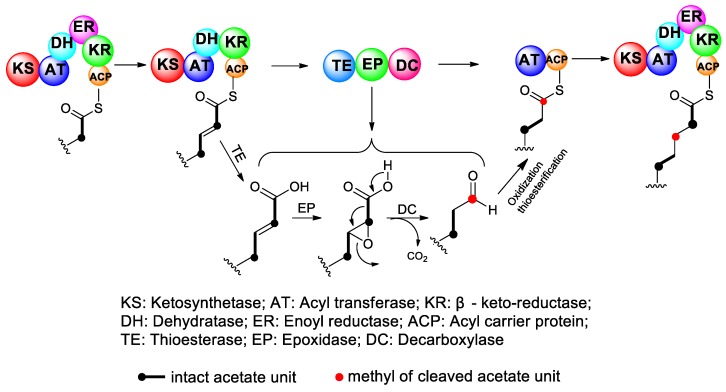
Proposed mechanism for the deletion of the carboxyl carbon from an intact acetate unit via specific functional modules within polyketide synthases (PKSs).

**Figure 5 marinedrugs-17-00594-f005:**
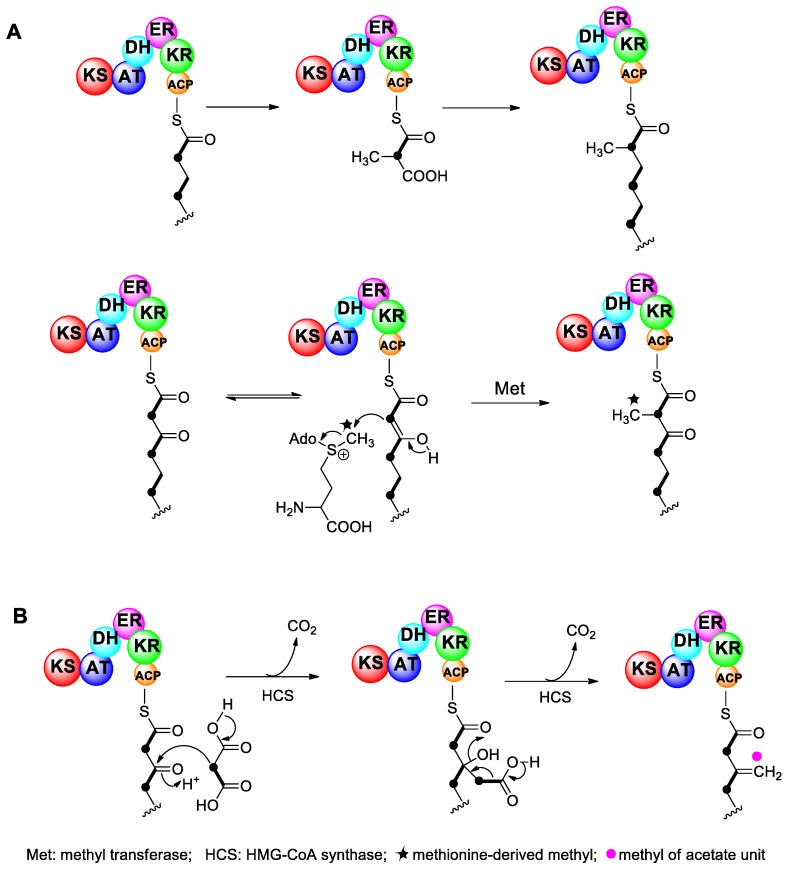
Two mechanisms of pendant alkylation of polyketides. (**A**) The upper reaction uses methylmalonic acid as a building block, while the lower one shows a methyl group derived from SAM. (**B**) A proposed mechanism for the introduction of a methyl side chain from a malonate unit via an HMG-CoA based enzyme.

**Figure 6 marinedrugs-17-00594-f006:**
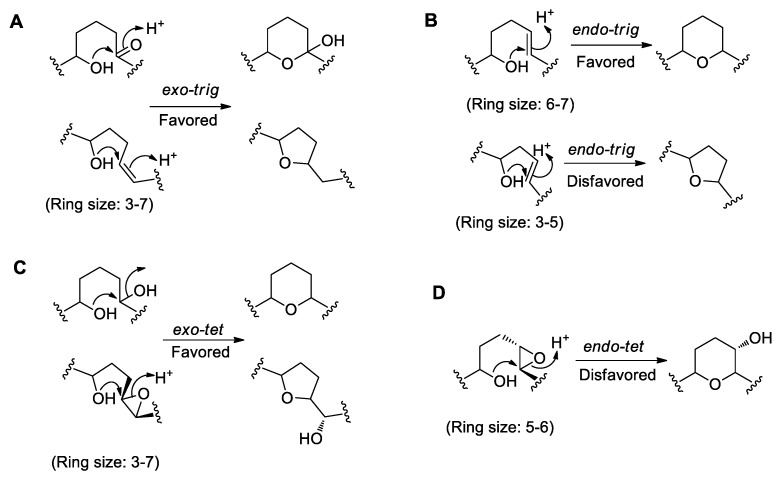
Mechanisms of ether ring formation relevant to dinoflagellate polyethers according to Baldwin’s rules. (**A**) *Exo-trig* cyclization of a hydroxyl group to a carbonyl group or a double bond. (**B**) *Endo-trig* cyclization of a hydroxyl group to a double bond. (**C**) *Exo-tet* cyclization of a hydroxyl group to a hydroxyl group or the opening of an epoxide. (**D**) *Endo-tet* cyclization of the opening of an epoxide.

**Figure 7 marinedrugs-17-00594-f007:**
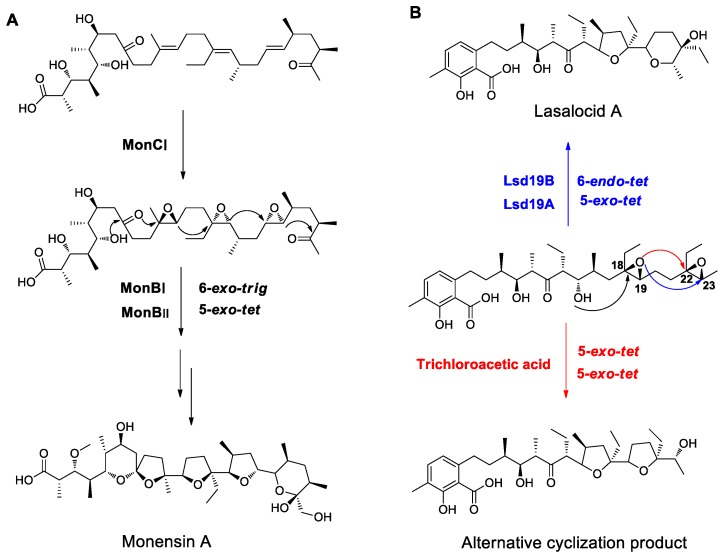
(**A**) Shows the mechanism of monensin A polyether ring formation. (**B**) Shows the mechanism of lasalocid A polyether ring formation. Enzyme-catalyzed cyclization produces the disfavored *endo-tet* product (up, blue) whereas acid-catalyzed cyclization produces the favored *exo-tet* product (below, red).

**Figure 8 marinedrugs-17-00594-f008:**
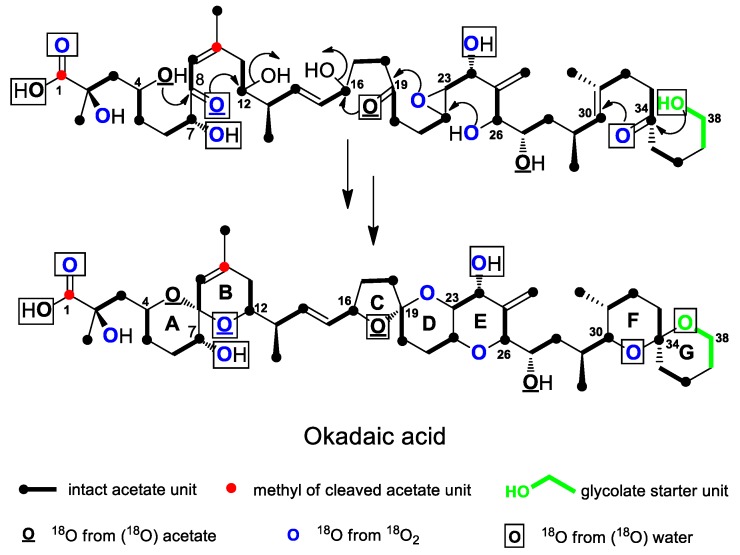
Possible mechanism for the ether ring formation in okadaic acid (OA) based on several isotopic labeling studies using (^18^O)-labeled acetate, O_2_, and water.

**Figure 9 marinedrugs-17-00594-f009:**
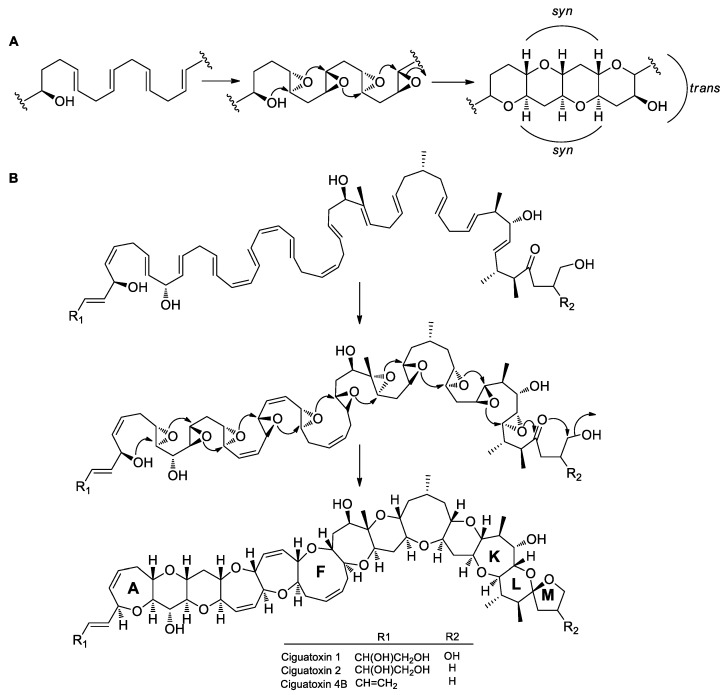
Possible mechanism of ether ring formation in fused-ring polyethers. (**A**) The *syn–trans* pattern of fused polyether ring formation. A polyene produced by PKS undergoes epoxidation and epoxide cyclization to form a fused polyether. (**B**) Proposed mechanism of ciguatoxins (CTXs) polyether ring formation.

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
