# Peer review of "Research Progress in the Biosynthetic Mechanisms of Marine Polyether Toxins"

_marinedrugs, 2019, doi:10.3390/md17100594_

Round 1
Reviewer 1 Report
The authors propose herein an interesting review on the metabolic pathways of polyether toxins produced by marine dinoflagellates. This is a hot topic that will be of interest for a broad range of scientists ranging from natural product chemists, molecular biologists, pharmacologists but also toxicologists so it clearly deserves publication in Marine Drugs. Overall the text is well written and easy to read.
As major change I would just say that the authors could structure a bit more their review by including some subparts in each part.
I would suggest some minor changes:
as soon as there is a transformation the authors should use the terms scheme. In Scheme 3 4 and 5 they should pay more attention in the mechanisms and arrows representing movement of electrons. Some of the transformations do not seem perfectly correct like in scheme 5 part B. For figure 6 some protons are missing as it is strange to see a nucleophilic oxygen attacking a nucleophilic double bond Some arrows are missing in figure 9Author Response
Dear Reviewer:
Thank you for your letter and for the comments concerning our manuscript entitled “Research Progress in the Biosynthetic Mechanisms of Marine Polyether Toxins” (ID: marinedrugs-614808). Those comments are all valuable and very helpful for revising and improving our paper, as well as the important guiding significance to our research. We have studied the comments carefully and have made corrections which we hope meet with approval. Revised portion were highlighted in the paper using the "Track Changes" function. The main corrections in the paper and the responds to the reviewer’s comments are as flowing:
Responses to the reviewer’s comments:
1. Comment: “As major change I would just say that the authors could structure a bit more their review by including some subparts in each part”.
Response: We do think this is a good suggestion, which will make our paper more clear and well-organized. We have divided the 2th part - Carbon Skeleton Deletion into three subparts, divided the 3th part - Pendant Alkylation into three subparts, divided the 4th part - Polyether Ring Formation into four subparts, and divided the 5th part - Gene Mining into two subparts, which we would hope will meet with approval.
2. Comments: “as soon as there is a transformation the authors should use the terms scheme. In Scheme 3, 4, and 5 they should pay more attention in the mechanisms and arrows representing movement of electrons. Some of the transformations do not seem perfectly correct like in scheme 5 part B. For figure 6 some protons are missing as it is strange to see a nucleophilic oxygen attacking a nucleophilic double bond. Some arrows are missing in figure 9.”
Response: we have rechecked the schemes carefully and consulted other literatures and professional books relevant to our paper, so as to ensure our schemes representing normatively and correctly, especially the electrons movement, protons transformation, and arrows orientation. Therefore, we redraw the Scheme 1, 3, 4, 5, 6, and 9, including the necessary arrows and protons, which we hope will meet with approval.
Once again, Special thanks to you for your good comments and suggestions.
Reviewer 2 Report
see attached file
